# Needs of Lung Cancer Patients Receiving Immunotherapy and Acceptance of Digital and Sensor-Based Scenarios for Monitoring Symptoms at Home—A Qualitative-Explorative Study

**DOI:** 10.3390/ijerph19159265

**Published:** 2022-07-28

**Authors:** Milena von Kutzleben, Jan Christoph Galuska, Andreas Hein, Frank Griesinger, Lena Ansmann

**Affiliations:** 1Division for Organizational Health Services Research, School of Medicine and Health Sciences, Carl von Ossietzky University of Oldenburg, Ammerlaender Heerstr, 140, 26129 Oldenburg, Germany; jan.christoph.galuska@uol.de (J.C.G.); lena.ansmann@uol.de (L.A.); 2Division for Assistance Systems and Medical Technology, School of Medicine and Health Sciences, Carl von Ossietzky University of Oldenburg, Ammerlaender Heerstr, 140, 26129 Oldenburg, Germany; andreas.hein@uol.de; 3Department of Hematology and Oncology at the Pius-Hospital Oldenburg, Georgstraße, University Department Internal Medicine-Oncology, 12, 26121 Oldenburg, Germany; frank.griesinger@pius-hospital.de

**Keywords:** lung cancer, immunotherapy, side effects, adverse events, patient’s needs, preferences, technology acceptance, palliative care, content analysis, qualitative research methods

## Abstract

Background: The development of immunotherapy in the treatment for lung cancer has changed the outlook for both patients and health care practitioners. However, reporting and management of side effects are crucial to ensure effectiveness and safety of treatment. The aim of this study was to learn about the subjective experiences of patients with lung cancer receiving immunotherapy and to explore their potential acceptance of digital and sensor-based systems for monitoring treatment-related symptoms at home. Methods: A qualitative-explorative interview study with patients with lung cancer (*n* = 21) applying qualitative content analysis. Results: Participants had trouble to classify and differentiate between symptoms they experienced and it seemed challenging to assess whether symptoms are serious enough to be reported and to figure out the right time to report symptoms to health care practitioners. We identified four basic needs: (1) the need to be informed, (2) the need for a trustful relationship, (3) the need to be taken seriously, and (4) the need for needs-oriented treatment concepts. The idea of digital and sensor-based monitoring initially provoked rejection, but participants expressed more differentiated attitudes during the interviews, which could be integrated into a preliminary model to explain the acceptance of digital and sensor-based monitoring scenarios. Conclusions: Supporting lung cancer patients and their health care providers in communicating about treatment-related symptoms is important. Technology-based monitoring systems are considered to be potentially beneficial. However, in view of the many unfulfilled information needs and the unsatisfactory reporting of symptoms, it must be critically questioned what these systems can and should compensate for, and where the limits of such monitoring lie.

## 1. Introduction

The development and rapid implementation of immunotherapy agents in treatment protocols for patients with non-small cell lung cancer has changed the outlook for both patients and health care practitioners [1]. Immunotherapy supports the body’s own immune system in recognizing and thus fighting tumor cells. While these promising novel treatment options are associated with great hope, at the same time, they confront patients and physicians with often complex side effects of an auto-immune nature that are uncommon in standard chemotherapy and require special clinical support [2,3].

Overall, immunotherapy has a more favorable side effect profile than comparable chemotherapy [4]. However, immunotherapies may cause inflammation, tissue damage, and misdirected immune activation in many different tissues and organ systems [5]. Adverse events range from fatigue to dermatological symptoms such as exanthema and pruritus, and gastrointestinal side effects such as diarrhea and colitis. In addition, endocrine, hepatic, and/or neurological side effects may occur. A rare but potentially life-threatening side effect of immunotherapies is immune-mediated pneumonitis [3,6]. With timely recognition and management, most of these side effects are treatable and reversible [7]. Given the increasing ambulatory nature of oncology care, practitioners rely on patient feedback to monitor adverse events [8]. This requires a high degree of personal responsibility and reflection from lung cancer patients receiving immunotherapy on an out-patient basis in dealing with a potential range of unspecific symptoms [9], and also highlights the need to educate patients comprehensively about their therapy and potential adverse events. Failure to recognize or report side effects can significantly compromise treatment safety and efficacy. Therefore, a systematic assessment and monitoring is essential, as stated in the German S3-Guideline Supportive Therapy in Oncology Patients [10].

### 1.1. Experience of Immunotherapy and Reporting of Therapy-Induced Symptoms in Lung Cancer Patients

There are already numerous empirical insights into the experience of illness and therapy in lung cancer under conventional therapy regimes such as chemotherapy. For patients and their families, the diagnosis of lung cancer usually means a fundamental crisis in their lives, which from then on is shaped by permanent uncertainty regarding the diagnosis, the therapy, and the illness trajectory [11,12].

So far, few data are available on the experience of and life with immunotherapies in patients with advanced lung cancer. In a qualitative study by Park et al., the experience of immunotherapy was divided in two phases: (1) the “avalanche”, a period of intense frustration and disorientation in search of effective treatment following diagnosis in which the disease determines large parts of one’s life and (2) the “living longer” phase, a timeframe characterized by a steady treatment plan without progression or adverse events in which components of the old life can be regained. Overall, immunotherapy was described as a positive experience, but also means a living in “limbo” and “cycling” between the two phases [13]. Many patients express a strong need for adequate information throughout the entire course of the disease and therapy. Therefore, providing information about diagnosis, the course of therapy, and the management of side effects should be understood as a structural aspect of care accompanying the disease owing to the continuous need among patients with lung cancer [14,15,16].

The reporting behavior of patients regarding symptoms is crucial in terms of effectiveness and safety of treatment, and patient’s quality of life. Nonetheless, communication about side effects is regularly characterized by deficits and incongruities [17,18].

Lung cancer patients receiving immunotherapy represent an empirically blind spot with regard to reporting of side effects. A recent qualitative study identified reasons for not reporting symptoms. These included firstly a perception that the experienced symptoms were not severe enough; secondly, an uncertainty about whether symptoms were actually side effects, and thirdly, the patient’s own expectation to manage symptoms without assistance. Fear of having treatment discontinued was mentioned but was not a prominent reason. The most common reasons for reporting symptoms were to ascertain if these were normal and expected, and to let health care practitioners know [19].

Another reason for not reporting side effects is presumably that patients often find it difficult to decide to contact their health care practitioners between regular appointments. As a result, symptoms must be reconstructed retrospectively at the next regular appointment. As a result, the therapy scheme may not be adjusted, or discontinued in time in the case of more serious side effects [8,20].

### 1.2. Improving Quality and Safety of Immunotherapy for Patients with Lung Cancer through Digital and Sensor-Based Monitoring Scenarios at Home

In a meta-analysis, the side effect-related therapy discontinuation rate under immunotherapy in patients with lung cancer and melanoma was reported to be 4.5% [2]. The incidence of side effects under immunotherapies is, on average, higher when they were combined with chemotherapy [4,21], which is the case for the majority of lung cancer patients. Due to the fact that the specific side effects of immunotherapies, especially when administered as part of a combination therapy, are often difficult to assess by the patients themselves [19], real-time monitoring can contribute to a timely reporting of side effects [22,23]. Furthermore, such an approach could counteract the phenomenon of “selective reporting” [18] by health care practitioners and patients.

One approach to improve safety and effectiveness of treatment could be digital [24] and sensor-based [25,26] monitoring of treatment-related symptoms at home. Specific measurements of biosignals (e.g., fever, skin color, respiration rate measured with wearables) and information on general behavior in everyday life (e.g., physical activity) measured with ambient systems could be used to complement subjective experiences and reports from patients by objective data. This could support patients in evaluating and monitoring their symptoms, provide health care practitioners with a more comprehensive picture, and help to detect critical symptoms in a more timely manner [24]. However, digital and sensor-based monitoring is not available in routine oncology care for lung cancer patients being treated with immunotherapy in Germany, and there is almost no empirical evidence on patients’ attitudes and acceptance of such approaches.

It is known, from other research contexts on the acceptance of technical applications in health care, that above all, the individually perceived benefit is a decisive criterion for (potential) users as also illustrated by the Technology Acceptance Model (TAM) [27] or its further development, the Unified Theory of Acceptance and Use of Technology (UTAUT) [28]. The UTAUT is widely applied in health care research. However, we considered the model with reservations due to the fact that important categories such as privacy and data protection [29,30,31] are not explicitly represented in these models.

### 1.3. Research Aims and Questions

The aim of this study was to learn about the subjective experiences of patients with lung cancer receiving immunotherapy, and to explore their attitudes and potential acceptance of digital and sensor-based systems for monitoring treatment-related symptoms at home. The study was guided by the following research questions: (1) How do patients with advanced lung cancer experience care and how do patients report symptoms? (2) What are their needs regarding treatment and interaction with health care practitioners? (3) What is the attitude of patients with lung cancer towards digital and sensor-based systems for monitoring treatment-related symptoms at home and what is the potential acceptance of such scenarios?

## 2. Materials and Methods

In view of the very limited empirical evidence, this study was designed as a qualitative-explorative interview study. The aim was to gain insight into the subjective experience of patients being treated for advanced lung cancer with immunotherapy, sometimes in combination with conventional cancer therapies and their attitudes towards a potential digital and sensor-based monitoring of treatment-related symptoms at home. The study was carried out by a multidisciplinary research team (public health/health services research, medicine and engineering) in a certified lung cancer center at the Pius-Hospital Oldenburg, Germany.

### 2.1. Recruitment and Sampling

All participants included in this study were treated for non-small cell lung cancer, except for one patient with small cell carcinoma. The following inclusion criteria were defined: (1) minimum age of eighteen years, (2) a confirmed diagnosis of advanced lung cancer (small cell or non-small cell), (3) knowledge of the German language to a sufficient extent to be able to be interviewed in German, (4) sufficient physical, emotional, and cognitive resources to be able to express oneself verbally in the context of an interview, (5) receiving immunotherapy or conventional forms of treatment to add a comparative perspective.

We aimed at a sample size of about twenty participants, which is in accordance with recommendations for qualitative interview studies [32], in order to capture a comprehensive range of lived experiences while keeping the amount of data manageable for an in-depth qualitative analysis.

In preparation for the study and the recruitment of study participants, the two researchers responsible for data collection (M.v.K. and J.C.G.) participated in the head physician’s weekly rounds at the hospital and in consultation sessions. Furthermore, the psycho-oncologist of the hospital was consulted. J.C.G. attended one week in the in-patient section of the lung cancer clinic.

At the beginning of the study, recruitment of participants took place in the context of out-patient consultation sessions. Patients were introduced to the study and its aims and methodology at the end of those sessions by the physician in charge (F.G.), and at least one of the two researchers, M.v.K. or J.G. Patients received a patient information sheet and were asked for their informed consent to be contacted by the study team. As both researchers experienced the consultation sessions as highly emotional and burdensome for the patients, the recruitment strategy was adapted to a postal approach. The lung cancer center identified potential participants fitting the inclusion criteria, and sent an invitation letter including the study information and consent forms to the patients inviting them to contact the study team. As a consequence of difficult recruitment (finding patients who met the inclusion criteria and consent), we had to apply a convenience sampling strategy.

All consenting participants received a phone call by the research team (M.v.K. or J.C.G.) to address open questions. If patients further agreed to participate, a date and place of their choice were scheduled. Patients could choose to be visited at home or to be interviewed at the hospital or at the university.

### 2.2. Data Collection

Most interviewees (*n* = 14 out of 21) chose to be interviewed at home, one interview took place at the University of Oldenburg, and six at the lung cancer center. Based on the current state of research, clinical experience of F.G., and categories from the UTAUT model [28], we developed a semi-structured interview guide that could be adjusted to the experiences of patients with or without prior experience with immunotherapy (see Table 1). In line with existing qualitative research [19] on the experience of immunotherapy in patients with lung cancer, we did not expect patients to be able to attribute symptoms accurately to their disease or its treatment. Therefore, we used “symptoms” as an umbrella term for any discomfort during or between treatment cycles. The interview guide was discussed with the psycho-oncologist of the participating cancer clinic and minor adjustments have been made. A pretest of the interview guide was conducted with three patients but no changes have been necessary.

To operationalize our second research question and to help the participants to gain a better understanding of different monitoring solutions, we developed three scenarios for monitoring common major side effects of immunotherapy: (1) insomnia and coughing [25,26], (2) fatigue and dyspnea [33,34], and (3) diarrhea [35] (see Figure 1) with the support of our colleagues from the division for assistance systems and medical technology (A.H.).

### 2.3. Data Analysis

Interviews were audio-recorded and transcribed verbatim. Interview transcripts were analyzed by structuring qualitative content analysis according to Kuckartz [36] supported by the software package MAXQDA (version Analytics Pro 2018).

The aim of the analysis was to structure, reduce, and analyze the data regarding the three main research interests of the study: (1) subjective experience of the therapy regimen with a focus on immunotherapy, (2) patients’ expectations and needs regarding therapy, and (2) attitudes and potential acceptance of monitoring treatment-related symptoms at home. Data analysis was carried out in the following steps:First review of the material: writing memos (to save ideas, concepts, hypotheses, and open questions) and case summaries including a summary of important clinical data (in order to be able to put findings and their interpretation into a clinical context).Deductive development of a basic coding framework reflecting the three main research interests and topics of the interview guide.Further differentiation of the coding framework and development of subcategories through inductive coding.(Re-)coding of the whole material with the final coding system.Interpretation of results within and between categories and development of a theoretical model of influencing factors and mechanisms to explain the acceptance of digital and sensor-based monitoring scenarios at home.

The subcategories were developed with regular discussion loops between J.C.G. and M.v.K. This served to check the plausibility and consistency of the categories. In case of dissent, the categories were discussed again until a consensus could be reached; where necessary, adjustments were then made to the category system.

### 2.4. Ethics and Other Permission

The study was examined and approved by the medical ethics committee of the University of Oldenburg (2019-035 date of approval 25 April 2019). All participants gave their written consent.

## 3. Results

### 3.1. Sample Characteristics

In total, 21 interviews were conducted and analyzed. In *n* = 9 cases, a partner or family members of the patient were present. Interviews lasted between 29 and 120 min. Main characteristics of the study sample are summarized in Table 2. Overall, the sample turned out to be heterogeneous regarding the treatment they were receiving or had received. N = 13 participants had received immunotherapy already. Overall, therapeutic schemes were by no means homogenous, and most patients (had) received different or a combination of treatments. In one case, immunotherapy had to be stopped due to massive therapy-induced diarrhea. Patients without experience with immunotherapy received other forms of cancer treatment such as chemotherapy, but immunotherapy was planned or considered as a further therapeutic option. Only one participant had not received any form of cancer treatment at the time of data collection.

### 3.2. Subjective Experience of Immunotherapy

Overall, patients experienced immunotherapy as relatively gentle with few side effects while, in contrast, chemotherapy is described as “pure poison” (16mT, Z. 34–36):


*“I also had chemotherapy. Of course I couldn’t take it too well, that’s quite clear, but no one can. Who can cope with that [chemotherapy]? … With immunotherapy, it’s not like that, I guess it’s not such a big deal.”*
(1mT, l. 88–92 & 456)

This positive attitude seemed more distinct if patients had received chemotherapy in the past. Owing to multiple combination, it was not always possible to relate certain experiences to a specific treatment.

#### 3.2.1. Sub-Category: Experience and Interpretation of Symptoms

Participants reported several symptoms, differing in timing, impact, and severity such as fatigue, pain, changes in appetite and sense of taste, changes in mucosal tissue and skin, pruritus, nausea, weight gain or loss and insomnia. Yet, it was often difficult for them to differentiate between tumor- or treatment-related symptoms: *“Now it’s all getting mixed up a bit*.” (2oT, Z. 4). Some patients experienced a dynamic development of symptoms forcing them to adapt their daily routines, but also giving them a sense of hope, as they saw a general possibility of improvement in this dynamic. Some patients identified familiar patterns and a certain regularity in the symptoms experienced during immunotherapy based on their prior experience with chemotherapy:


*“I also think that the side effects are always the same, just like with chemo. The first day this, the second day that and at some point you’re through and then you start all over again. At least that’s how it was with the chemo.”*
(4oT, Z. 912)

This helped them to find a sense of coherence during treatment and contributed to a general positive expectation towards therapy-related symptoms *“when you recover, then it’s all good again.” (1mT, Z. 463)*

At the same time, symptoms that occurred for the first time or are perceived as severe or even potentially dangerous provide a serious challenge and burden for patients receiving immunotherapy. Patients find themselves in a situation where they are self-reliant in handling and assessing their symptoms, and are often unsure whether to seek immediate medical advice:


*“Yes, the question was last time. I had a bit of constipation and then I had a fever. And then I didn’t know if I should go to hospital, because they said that if I went over thirty-eight [degrees Celsius], I should go to hospital.”*
(6mT, Z. 1172–1175)


*“Sometimes we stand here a bit foolishly. Well, we didn’t know whether to call an ambulance or not.”*
(10mT, Z. 1115–1117)

Especially regarding symptoms related to immunotherapy, patients reported unmet information needs regarding their ability to differentiate treatment-related symptoms from symptoms occurring for reasons unrelated to the cancer or its treatment. In one case, for example, the immunotherapy had to be stopped due to excessive immune-mediated diarrhea that was initially and erroneously (self-) diagnosed as food poisoning by the patient, the family, and the general practitioner:


*“Because no one told us anything before, there was nothing in the papers. And so the side effects started with diarrhea. I didn’t know that at all. … the nurse always asked do you have diarrhea? I thought, why is she asking?” [Interviewer asks about the content of the information session for immunotherapy] They didn’t say anything about the side effects.”*
(4mT, line 164–176)

In some cases, certain symptoms such as coughing and dyspnea had accompanied patients for a long time before the lung cancer diagnosis and treatment. Patients, therefore, experienced difficulties in assessing the severity in the context of cancer treatment. Another phenomenon influencing their interpretation was the habituation to unspecific symptoms such as fatigue.

#### 3.2.2. Sub-Category: Awareness and Reporting of Symptoms

Our results demonstrate that patients do not necessarily perceive symptoms as such. For example, one patient, whose general state of health as well as her emotional situation were obviously impacted by the cancer and its treatment, talked about various symptoms during the interview. However, being asked explicitly whether she had symptoms that could be monitored with digital and sensor technology at home, she denied it, arguing: *“Because I’m fine … nothing is wrong with me… I am healthy*.” *(6mT, 1330–1334).*

In general, participants considered it as important to inform their health care practitioners at the cancer center about symptoms that appeared during treatment. Some even believed it to be their obligation as a patient:


*“No, theoretically I should get the cudgel, if I didn’t mention things I noticed that might be related to the therapy to the doctor.”*
(7oT, Z.127)

Fear of severe complications resulting from symptoms not treated in a timely manner was the major reason for patients to report symptoms. Only one patient expressed the concern that due to her reporting of symptoms, treatment might be discontinued. Another reason for non-reporting was the lack of a trustful patient–physician relationship:


*“I didn’t say anything. I didn’t even know the doctors. What should I do with the doctors if I don’t know them?”*
(1mT, Z. 324)

Other participants assumed that many patients had doubts whether the symptoms were sufficiently significant and therefore hesitated to report them. In view of the severity of their disease and the complexity of its treatment, some patients did not consider themselves as competent interaction partners for their health care practitioners. This self-perception might be another reason holding patients back from discussing symptoms with their health care practitioners. Another phenomenon was the attribution of a symptom, fatigue in this case, as part of one’s character rather than as a treatment-related symptom which therefore remained unreported:


*[Interviewer]: “With the fatigue, you didn’t mention that?”*



*[Interviewee]: “Nope, because that’s too much because of myself. I think. Because I’m so listless, because I’m so lazy.”*
(6mT, Z. 392)

### 3.3. Patients’ Needs Regarding Treatment

The analysis of the patients’ narratives yielded four basic needs expressed related to their cancer therapy and communication with their health care practitioners. These needs seem to be important for all patients regardless of the specific treatment and the symptoms they experienced.

#### 3.3.1. Sub-Category: The Need to Be Informed → Comprehensive Information and Continuity of Information

Irrespective of the type of treatment they received, patients expressed a strong need for information about their disease, its treatment, and what to expect in terms of logistics and possible side effects:

*“…but first priority is talking to the doctors first. That is important. That you are fully informed. What are the side effects? What about the radiation? What gets broken there and how does it work with the chemo?”*.(4oT, l. 426)

Many participants treated on multimodal protocols with multiple treatments expressed unmet information needs in regard to treatment and related symptoms. The initial consultation session is considered to be central for patient information. It is commonly accompanied by strong emotions such as life-threatening fear or a feeling of being in shock, while most patients have no experience with the disease and its treatment options. Patients often described the consultation sessions as a rapid procedure, leaving them with a perceived lack of understanding about the next steps and what to expect from the therapy they will be receiving:


*“I mean I don’t know the disease. And you don’t have any experience with it, you can’t rely on previous experiences. And you simply have to be informed a bit more comprehensively and then also taken seriously.”*
(7mT, Z. 291–293)

Taking enough time and including relatives in the counseling sessions was valued as extremely helpful and was practiced by some physicians, but also missed by many patients in our sample. Patients suggested to improve quality of information by giving patients the opportunity to contact their health care practitioners on a regular basis between treatment cycles.

Moreover, patients expressed a need for continuity of information related to the initial counseling and follow-up sessions. Patients reported that they had received different and sometimes contradictory information and diagnostic results from different health care practitioners (e.g., the oncologist and the radiation oncologist). Consequently, they sometimes went through emotionally burdensome situations or felt surprised and unprepared for the next steps in their therapy circle.

#### 3.3.2. Sub-Category: The Need for a Trustful Relationship with Health Care Practitioners → Respectful Dealings and Continuity in Interpersonal Relations

Patients valued respectful and sensible communication. However, negative experiences such as *“petulant”, “snotty” (7mT, l. 517; 537)* or aggressive behavior of physicians were reported, which had a significant impact on patients’ overall wellbeing, satisfaction, and trust in their physicians:


*“I was lying on the thing and he said to me, “Where are your blood results?” I had already handed in all my blood results. I had brought everything with me and handed it in. And then I still had some in my pocket and he shouted at me. Ooh, he shouted at me: “You must know where your blood values are and you must [know your blood values]!” And I didn’t know that… I never had to. I never needed to know my blood values … and then I gave them to him again and then he ran off, but I think he totally freaked out.”*
(1mT, line 362–364)

Patients expressed the need for a contact person as they perceived treatment and communication to be fragmented between the multidisciplinary team in the clinic which makes it difficult to build up trustful relationships.

Moreover, patients described that not all symptoms occurring with the cancer treatment can be handled by the lung cancer center. Thus, patients see other specialists who are often not prepared to treat patients with lung cancer and especially lack knowledge about immunotherapy. Furthermore, some patients feel overchallenged by finding the right specialist for their problem. Patients therefore would find it helpful to be provided with a list of recommended specialists with additional knowledge and experience in treating patients with lung cancer.

#### 3.3.3. Sub-Category: The Need to Be Taken Seriously with Subjective Symptoms → Adequate and Timely Reaction to Expressed Symptoms in Every Setting

Patients expected physicians to explicitly ask about symptoms they might be experiencing and to give a differentiated explanation. A question about one’s wellbeing and a global explanation of symptoms is perceived as insufficient:


*“And when I go to the doctor and tell him that I do not feel well and he says: well, that is because of the chemotherapy, than this does not help me further.”*
(7mT, Z. 176–180)

In fact, this participant had been referred to the intensive care unit shortly afterwards due to massive dyspnea and very concerning blood values. This patient felt that he was neither asked adequately about symptoms, nor did he receive an adequate response to his description of perceived symptoms aimed at averting possible danger or harm to his health.

Patients see physicians on duty of an explorative and committed counseling behavior sensitive to detecting possible treatment-related side effects in the patients’ narratives. This was particularly important in conversations at the phone, when patients called the clinic between treatment cycles when facing challenging situations at home. When patients are dealing with symptoms at home, counseling over the phone is assumed to be extremely supportive in giving advice whether to stay home or seek for immediate medical help. This might be of utmost relevance for patients receiving immunotherapy due to the unspecific spectrum of symptoms.

#### 3.3.4. Sub-Category: The Need for a Treatment Concept That Addresses Medical and Personal Needs on an Equal Basis → Perceived Coherence and Continuity of Treatment Pathways, Minimizing Avoidable Burden for Patients

Besides the need to be treated with respect and being taken seriously with regard to their symptoms, participants found it important to also improve patient centeredness in terms of the organization of care pathways. It seems essential for most patients to comprehend the structure and scheduling of their therapy. Long waiting times for appointments with specialists or for urgently expected diagnostic results, especially at the beginning of the therapy, are described as an enormous burden. Additionally, appointments during a visit to the lung cancer center should be well organized to avoid excessive waiting times and to reduce potential burden on patients. In single instances, patients described that they felt responsible for scheduling appointments on their own initiative, although they would have rather handed over all organizational responsibility to the medical team at the center. Likewise, it seems of equal importance and a sign of respect, if circumstances of cancer treatment do not only address medical needs, but are also tailored to the living situation, e.g., taking into account long traveling distances to the lung cancer center when scheduling and organizing appointments.

### 3.4. Attitudes and Potential Acceptance of Digital and Sensor-Based Monitoring Systems at Home

This theme reflects the attitude and potential acceptance of digital and sensor-based technologies to monitor symptoms in the patient’s home. Initially, the interviewees expressed an intuitively negative attitude and discomfort towards the idea of linking one’s own body or living environment with technical systems. Ambient systems in particular were viewed with skepticism, due to their potential for ubiquitous observation of the entire household. Most of the participants were unfamiliar with digital and sensor technology approaches in the health context. Potential benefits were not seen at first sight and the increasing mechanization in this area is perceived as *“exaggerated” (13mT, line 816).* In general, the interviewees believed that they could report symptoms and complaints themselves: *“That wouldn’t be for me. I have to say quite honestly, well, I can help myself” (11mT, Z. 568–570)* or considered such technical solutions only useful for people living alone. With a more intensive thematic discussion, this attitude tends to weaken in the course of the interviews.

#### 3.4.1. Sub-Category: Assumed Usefulness of Digital and Sensor-Based Monitoring Systems

The assumed personal usefulness of a technical monitoring system could be identified as the most important criterion participants applied in the interviews. Interviewees usually tended to be more open to the use of technology and the reasons for refusal were put aside, if the assumed usefulness, mainly defined through medical benefits, would be deemed as sufficiently significant in the individual case.

An improved assessment of symptoms and side effects by using a technical monitoring system was perceived as potentially beneficial. Many patients struggled to assess the severity of symptoms and would therefore find it supportive if the monitoring system gave appropriate feedback. One participant suspected that especially patients who had lived with symptoms for quite a while could benefit from such an external monitoring:


*“Especially for long-term patients, where the habit simply creeps in. It’s like driving a car when the brakes slowly deteriorate. People who drive every day don’t realize that. If someone else drives it, they say, “My God, how far can you push the brakes?” (…) And, if you are sick for a longer period of time or have a cough for a longer period of time or whatever, you don’t recognize it anymore … You just get used to it. And that’s the bad thing. Well. That you can no longer decide for yourself. Is it really better or worse now?*
(7mT, lines 1272–1300)

Participants expressed different attitudes towards a system that would be able to submit information to their health care practitioners. Some participants found it promising that technical solutions might enable them to receive out-patient cancer treatment while improving quality and safety of the therapy: *“It is most comfortable at home. But you’re often afraid that you’ll miss something”* (3mT, line 857).

Yet, others were concerned about their autonomy. A few interviewees assumed that monitoring throughout the night could be of extra benefit, as some symptoms (e.g., nocturnal coughing) might pass unnoticed while asleep. Furthermore, a monitoring system should be able to register changes in context and activity, and effects on measured parameters, as temperature inside their homes or outside during sport activities.

Another assumed benefit of technical monitoring was support in challenging situations. Related to the difficulties in deciding whether a perceived indisposition might be harmless or serious, participants saw a potential to prevent adverse events or an acute health crisis. Next to feedback from the system to the patient on how to react adequately, interviewees suggested to implement an emergency call function and found it essential that the technical monitoring included a solution for direct communication with their health care practitioners. Overall, the expressed needs regarding monitoring systems varied greatly depending on the individual situation and symptoms experienced in the past. Monitoring was perceived as particularly useful for symptoms that had been experienced in the past. Still, most participants felt that benefits of monitoring would not (yet) be great enough at the time of the interview, although stating that this could change in the future if their own health condition worsened. In general, a sensitive detection of an incipient emergency situation was seen as much more useful than an undirected continuous monitoring of the patient at home:


*“They [technical monitoring systems] don’t have to report every little stomach cramp, but when it’s life-threatening. Then they should take action … then I think it makes sense.”*
(12mT, line 919)

However, most participants considered these benefits mainly relevant for elderly patients or cognitively impaired patients being unable to express themselves adequately and/or patients with lung cancer living alone. Concerning their own situation, interviewees tended to rely on their ability of self-observation and on their families.

The need for a sense of security and reassurance was strongly evident in our sample. In addition to an objective improvement of the effectiveness and safety of treatment, interviewees saw the potential in digital and sensor-based monitoring systems to promote a patient’s confidence in interpreting and managing symptoms at home, and to improve the exchange between patients and health care practitioners between treatment intervals.

Interviewees saw some additional benefits from the use of digital and sensor-based monitoring systems above and beyond medical and psychological benefits. Monitoring of complaints can provide objective evidence of limited functional performance or suffering. This could help, for example, to prove one’s own illness-related limitations in a private, professional or medical context and thereby to defend themselves against excessive demands. Another benefit mentioned was that monitoring systems could collect data on processes that are otherwise difficult to quantify, e.g., how much a person moves or gets up at night. In consequence, monitoring could aid in making these processes objectively comprehensible. Additionally, potential users expressed a higher acceptance of home monitoring systems if the data collected would be useful for research and thus for the general public. It was argued that they would benefit from modern state-of-the-art therapies themselves and could contribute to scientific progress in return.

#### 3.4.2. Sub-Category: Arguments against the Use of Monitoring Systems

Even though participants showed a tendency to become more open towards technical monitoring systems during the interview, two major arguments could be identified against their use.

Lack of assumed usefulness was the main argument against it. Participants made a personal cost-benefit balance. A subjectively good health situation with few symptoms, a high level of confidence in one’s own self-observation competence, as well as doubts about the significance and practicability of the monitoring and alarm systems, had a diminishing effect on the assumed benefit. Moreover, many potential users believed attentive and caring relatives are *“the best sensor ever” (8mT, daughter, Z. 893)* which made a technical monitoring system unnecessary. It became apparent that the potential users and their relatives had great trust in their “human” perception and its superiority in terms of sensitivity and specificity compared to the technical measurements of a monitoring system:


*Relative: “But I could tell you that, whether she’s sitting or sleeping now because she’s overtired or exhausted or whether she’s just sitting there and…”*



*Patient: “brooding again.”*
(4oT, Z. 686–690)

Even though participants in our sample frequently reported difficulties in evaluating symptoms, they showed a tendency towards imagining monitoring options to be applicable to those symptoms or problems that are easy for them to interpret and report: *“Whereby you could also say that objectively yourself. I can’t get up the stairs any more”* (3mT, Z. 485).

Furthermore, the respondents suspected that the sensitivity of monitoring systems in detecting subacute and chronic but potentially dangerous courses (such as pneumonia) might not be high. The same applied to the specificity of the measurement. Here, for instance, the interviewees doubted that a pressure sensor attached to the bed could differentiate between heavy coughing and other activity resulting in false alarms.

Another argument against a technical monitoring at home were possible disadvantages of using monitoring technology. Some participants worried about being disturbed by the technical devices in their domestic environment due to sounds, exposure to radiation, and hassle if data had to be provided proactively by the user (e.g., in the case of online reporting of symptoms). A few patients were concerned that such systems might place the disease even more in the center of their lives, therefore making it difficult putting the disease and related threats aside in phases with little symptoms. Additionally, interviewees expressed the latent fear that their autonomy and self-determination could be violated by digital and sensor-based monitoring systems.

#### 3.4.3. Sub-Category: Concerns Related to Privacy and Data Protection

Issues of data protection and privacy turned out to be relevant and sensitive aspects for potential users of home-based technical monitoring systems. Interviewees expressed various concerns, despite being generally willing to take certain risks if the medical benefits were considered sufficiently large.

There is a pronounced need among participants to not become completely transparent. This refers not only to themselves, but also to the household and third parties staying there. Especially in the case of ambient monitoring systems, a strong intrusion into the personal rights of relatives or guests is feared. The idea of *“total surveillance” (2oT, Z. 744)* resonates with almost all interview partners.

The type of measurement technology plays a major role in the perception of privacy. While, for instance, the use of pressure sensors at the bedside still seems acceptable, video recordings are rejected. Even with a strong reduction of the resolution by subsampling of the recordings, such a form of monitoring would be out of the question for most of the participants. The location of the measurement is also of great significance for the respondents: ambient technologies in toilets and bathrooms are little accepted. Furthermore, many respondents found it important that they could deactivate the monitoring system. However, it is repeatedly emphasized that privacy and benefits must always be weighed against each other individually.

Another aspect that caused discomfort in terms of an invasion of privacy was the fear that such recordings could reveal deficits in one’s own health behavior or would even be able to map a character weakness: *“You don’t want them to see that you didn’t do anything.” (6mT, Z. 732).* This expresses the apprehension that the monitoring system could become a normative control authority in the private everyday life of the users.

The issue of privacy is closely related to concerns regarding data protection. Maintaining control over and protecting one’s data was a key priority for our participants. The respondents feared misuse of their data by institutions, e.g., their health insurance company. Monitoring was only potentially acceptable if the processes of data storage and processing were transparent. Codetermination on the location of data storage was essential and a local storage in the user’s home was preferred. Many of the respondents considered an automatic forwarding of their data in case of an emergency as desirable. However, it was crucial that any forwarding of data by the monitoring system would only happen with their explicit permission and only to a trustworthy individual in the responsible health facility:


*“But I am quite careful with data. You hear it all the time and you also see it on TV that patient things end up I don’t know where. And mine, only the doctors should get them (…) a computer doesn’t forget … Even if you say it’s entered and if I delete it, it’s gone, it’s not completely gone.”*
(16mT, Z. 649–667)

### 3.5. A Preliminary Model to Explain the Acceptance of Digital and Sensor-Based Monitoring Scenarios at Home among Lung Cancer Patients

As the last step of data analysis, the interpretation of results within and between categories, we integrated our results into a preliminary theoretical model to explain the acceptance of digital and sensor-based monitoring scenarios at home (see Figure 2). The model presents influencing factors and mechanisms in the development of attitude towards and therefore the acceptance of digital and sensor-based monitoring at home.

The assumed usefulness of the respective monitoring technology turned out to be the core category regarding attitude and acceptance towards digital and sensor-based monitoring at home. The assumed benefit is primarily defined by an improved assessment of therapy-related complaints and prevention of impending health crises. Both aspects meet the respondents’ need for safe treatment and care. The perception of potential individual usefulness of a domestic monitoring depends on individual characteristics. However, besides the potential benefits, there are numerous arguments against the use of monitoring technology. While initially, most participants showed a reserved attitude towards the idea of digital and sensor-based monitoring at home, they started to balance their reasons for rejection against the assumed usefulness considering their personal situation over the course of the interviews. It was assumed that in individual cases, the expected benefits could outweigh individual concerns. Finally, neutral factors could be identified that can influence the attitude and acceptance both positively and negatively. These include previous experience in dealing with technology and the general affinity of a person for technology as well as the attitude of relatives.

Attitude and acceptance, as it became clear in this study, are by no means to be understood as fixed outcomes, but are subject to the dynamic influence of various factors, e.g., the health situation and the experience of illness and therapy, experienced complaints, the individual social situation, but also the cognitive engagement with the idea of technology-based monitoring at home.

## 4. Discussion

In our study, we interviewed patients with lung cancer, who either had received immunotherapy or were considered for immunotherapy in the near future, about their treatment experience and reporting of symptoms. We reconstructed their needs regarding treatment and interaction with health care practitioners and proposed and discussed digital and sensor-based scenarios for the monitoring of common side effects of immunotherapy in lung cancer treatment. In line with previous qualitative research [19,22] immunotherapy in lung cancer meant a generally positive and promising outlook for our participants. At the same time, the reporting of potentially treatment-related symptoms to health care practitioners is not a trivial act: The participants in our sample had trouble to classify and differentiate between the symptoms they experienced. This is probably also because many of the participants had received a combination of different therapies. As in previous qualitative research [19], we did not find the one major reason, such as fear of having treatment discontinued, for non-reporting of symptoms. Instead, most of the participants seemed unaware of the range of side effects of immunotherapies and the health complications that potentially accompany them. In addition, it seems challenging for the patients to assess whether symptoms are serious enough to be reported and to figure out the right time to report symptoms to their health care practitioners.

As previous research has shown, the prevalence of unmet needs among cancer patients is significant and highest during treatment intervals [15]. Our study adds empirical insights on the needs of lung cancer patients related to immunotherapy. From the participants’ narratives on their experience of treatment at the lung cancer center, we were able to reconstruct four basic needs: (a) the need to be informed, (b) the need for a trustful relationship with health care practitioners, (c) the need to be taken seriously with subjective symptoms, and (d) the need for a treatment concept that addresses medical and personal needs on an equal basis. Across all these needs, continuity and coherence appeared to be central concepts for patients and many of the unmet needs occurred related to these concepts. However, participants in our sample mostly did not report (unmet) needs explicitly, but needs were rather latent in the data and were reconstructed by the qualitative analysis.

The fact that patients do not directly report unmet needs in connection with immunotherapy may be due to the nature of the therapy itself, among other reasons. The side effects of immunotherapies during treatment are far less impressive for patients than those associated with conventional chemotherapy [2,3,19]. This may also be one of the main reasons why symptoms are reported more hesitantly, or the necessity is only seen in the case of subjectively severe symptoms.

The results of our study demonstrate that, in principle, patients feel responsible for communicating complaints to their therapists. However, a central prerequisite for this is a trusting relationship with the practitioner and the feeling of being able to interpret and communicate medical complaints and symptoms competently. In line with other research on lung cancer, we found (unmet) needs related to continuity of information [14,37] as well as quality and continuity in the relationship with health care practitioners. Both might have an impact on both the frequency and the quality of patients’ symptom reports. A trustful relationship that is perceived as reliable over the course of treatment might be the key to comprehensive reporting of symptoms in immunotherapy. It has been stated before that there are discrepancies between cancer patients’ and health care practitioners’ interpretation and reaction to symptoms [18,21,34]. With regard to immunotherapy, this indicates that the feeling of being taken seriously and a timely reaction to expressed symptoms, even if they seem rather minor, is extremely important to improve safety and effectiveness of treatment as well as the wellbeing of the patients.

Previous research has already pointed out the importance of real-time monitoring of symptoms and adverse events during immunotherapy in cancer patients [22,23]. In our study sample, the idea of digital and sensor-based monitoring turned out to be completely new to most of the participants and initially provoked rejection. However, this attitude at least partially eroded in most participants during the interview when introduced in more detail to the scenarios where an effect has been found in other contexts as well [38].

In accordance with established models to explain technology acceptance such as the TAM [27] or UTAUT model [28], the perceived usefulness turned out to be the most important factor in the participants’ potential acceptance. Although these models are regularly applied in the context of technical applications in health care, they proved to be only of limited use to comprehensively explain the (potential) acceptance of digital and sensor-based monitoring in the context of our study. We found some aspects that are not explicitly named as relevant for acceptance in the cited models.

Promoting a sense of safety with regard to evaluating symptoms with the support of a sensor-based monitoring system was seen as a major potential benefit and therefore impacted acceptance. This was also found as an effect in a study on digital monitoring and management of symptoms in lung cancer patients receiving immunotherapy [24]. However, for potential users, the prevention of acute emergency situations, including the reporting of related data directly to their health care practitioners, is a priority rather than detection of treatment-related symptoms that might be a warning sign for treatment-related complications.

Like in other studies [30,31], issues of privacy and data protection turned out to be essential for the acceptance of monitoring symptoms at home. The need for trustful relationships became relevant in this context, too, as a direct linkage of the monitoring system to a personally known health care practitioner of trust would help reduce concerns about misuse, for example. At the same time, it became also evident that our participants were willing to weigh things up and that a high personal benefit also makes concessions to data protection conceivable.

An aspect that was unexpected to us and has not been described in previous research on this group of patients is the fear of normative control that turned up in our sample related to digital and sensor-based monitoring in one’s everyday life. Concerns were expressed that health care practitioners would be able to detect a lack of health-promoting behavior or even character weaknesses in their patients through the data delivered by a monitoring system.

Jamieson and colleagues found the phenomenon of “personalization” of symptoms, meaning that patients would benchmark any symptom against what they think would be “usual” or “typical” for them as a person [22]. However, this aspect seems to be still under-researched.

Another important factor seems to be the desire of lung cancer patients to maintain normality—“living life as usual” [11]—in the sense that the implementation of a monitoring system in the patient’s home was rather seen as something constantly challenging this endeavor rather than supporting it.

Respondents’ narratives evidenced various needs related to health care delivery during lung cancer treatment, such as reaching a contact person at the clinic, involving family members, or taking personal circumstances into account in the organization of treatment circles. At this point, a further starting point for supportive measures through eHealth solutions arises, which could be mapped to the requirements of potential users, e.g., through apps that allow intervention-specific information, help for self-help, information for relatives as well as means of asynchronous communication with the clinic.

### Limitations and Methodological Considerations

Overall, the qualitative explorative approach provided us with a rich set of data to answer our research questions, but we were facing several (methodological) challenges especially related to recruitment and sampling as well as in data collection. Our goal was to capture a comprehensive range of lived experiences in regard to our research questions. However, as recruitment turned out to be challenging, (a) for the pragmatic reason of finding enough patients receiving or being eligible for immunotherapy within the time frame of the study and (b) in terms of willingness to participate during a sometimes emotional and/or physically extremely challenging situation, we were not able to perform a purposeful sampling but rather had to be satisfied with a convenient sample. However, our sample is fairly representative of patients with lung cancer in an out-patient setting in terms of age, distribution of stages, sex, etc. Furthermore, as in clinical reality, very few patients receive immunotherapy as a single treatment modality, and our sample turned out to be very heterogeneous with respect to treatment. Therefore, it was challenging to collect data reflecting the experience of immunotherapy exclusively. In addition, the sample included mostly patients with a satisfactory health status with low subjective burden caused by treatment-related symptoms at the time of the interview. Many participants commented that they would find monitoring more useful if their health status declined. Additionally, it should be noted that all participants were recruited from one cancer center. Therefore, the experience of and needs regarding treatment and communication with health care professionals has been probably impacted by the context conditions of this specific center, its organization, and culture.

A particular methodological challenge in gathering attitudes toward technology-based home monitoring systems was communicating and visualizing the complexity of digital and sensor-based technologies, and their individual possibilities. It is questionable whether all participants received a concrete idea of the functions and possible advantages and disadvantages. Furthermore, it became clear that the interview per se meant an intervention in terms of changing acceptance as most participants adopted a more open and differentiated attitude in the course of the interview.

## 5. Conclusions

This study stresses the importance of finding ways to support lung cancer patients receiving immunotherapy and their health care providers in observing, reporting, and communicating about treatment-related symptoms. Overall, lung cancer patients, as potential users of digital and sensor-based monitoring systems, despite all reservations, do see potential benefits for themselves and others under certain circumstances and conditions. However, technical solutions are not regarded in isolation but rather provide a potential bridge between health care practitioners in the clinical setting and patients at home. In view of the many unfulfilled information needs and the unsatisfactory reporting of symptoms, however, it must be critically questioned what technology-based monitoring can and should compensate for and where the limits of such monitoring lie.

Further research is needed to better understand the information needs of patients with lung cancer before, during, and after being treated with immunotherapy. The development of monitoring systems based on patient-reported outcomes for these specific groups would be a conceivable approach. These could enable targeted monitoring, provide desired contact options to health care practitioners, and offer practical information on treatment-related symptoms, as well as care options. Such applications have already shown promising results in some studies [39,40,41].

## Figures and Tables

**Figure 1 ijerph-19-09265-f001:**
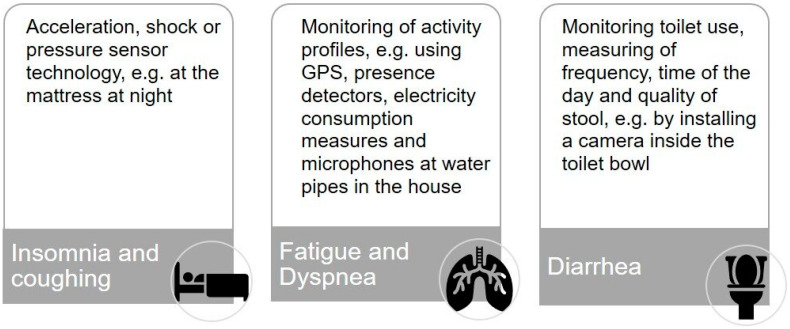
Digital and sensor-based scenarios for monitoring treatment-related symptoms of immunotherapy in lung cancer at home.

**Figure 2 ijerph-19-09265-f002:**
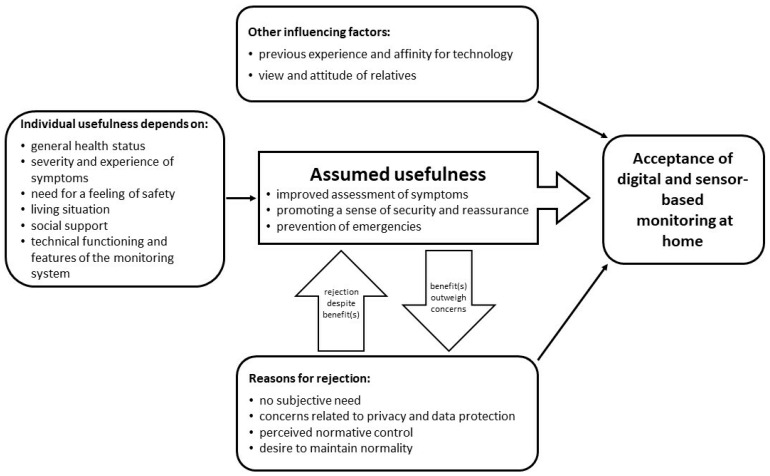
A preliminary model to explain the acceptance of digital and sensor-based monitoring scenarios at home among lung cancer patients.

**Table 1 ijerph-19-09265-t001:** Structure and content of the interview guide.

Part 1: Experience of the disease and its treatment	Introduction: Narrative-generating initial question on general feeling and experience of therapy
	Experience of and coping with symptoms
	Perception of the immunotherapy education talk Communication with health care professionals Coping and needs in everyday life
Part 2: Attitude towards and potential acceptance of digital and sensor-based monitoring systems at home	Introduction of the idea of monitoring treatment-related symptoms at home Potential usefulness and benefits of monitoring from the user’s perspective Introduction and discussion of different monitoring scenarios (see Figure 1) User participation and data protection or privacy-related issues Expectations of monitoring systems and concerns from the perspective of potential users

**Table 2 ijerph-19-09265-t002:** Sample characteristics.

Participants		*N* = 21
Sex (%)		14 male (67%) 7 female (33%)
Age (in years)	Median	65
	Range	50–81
Histological type	Adenocarcinoma	16
	Squamous cell carcinoma	4
	Small cell carcinoma	1
Stage	II	1
	III	3
	IVA	7
	IVB	9
	Stage unclear	1
Therapy goal	Curative	3
	Palliative	17
	Unclear	1
Therapy Scheme	Mono-IT *	2
	Mono-IT *, status post RT/CT/RCT	6
	Combination treatment * (IT + RCT/CT, vaccination if applicable)	4
	Status post IT + RCT *	1
	RET-Inhibitor (BLU−667)	1
	RCT	1
	Status post RCT, RT and/or surgery, IT-scheme planned if applicable	5
	No treatment received yet	1
Living situation	Living alone	2
	Living with partner or family	18
	Other arrangement	1

Abbreviations: IT = Immunotherapy; TT = Radiotherapy; CT = Chemotherapy; RCT = Radi-ochemotherapy; * patients with immunotherapy experience (total *n* = 13).

## Data Availability

The anonymized data presented in this study are available on request from the corresponding author, if participants gave their consent for data sharing.

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
