# Peer review of "Needs of Lung Cancer Patients Receiving Immunotherapy and Acceptance of Digital and Sensor-Based Scenarios for Monitoring Symptoms at Home—A Qualitative-Explorative Study"

_ijerph, 2022, doi:10.3390/ijerph19159265_

Round 1

Reviewer 1 Report

Thank you for the opportunity to review your paper, which discusses “Needs of lung cancer patients receiving immunotherapy and acceptance of digital and sensor-based scenarios for monitoring symptoms at home.”

 Overall, the paper is well written and provides a good overview of a well-designed study. Findings are clearly presented and there is a clear interpretation of study results.

I will provide some comments under each heading of the paper as to where content could be strengthened, or where edits are required.

Abstract

The results reported in the abstract relate to only one section of the results reported in the main body of the paper (section 3.3). It would be good for the authors to also add some results related to section 3.2 (patients’ experiences of immunotherapy including awareness and reporting of symptoms) and section 3.4 (attitudes and potential acceptance of digital and sensor-based monitoring systems at home). By reporting on the findings of these two sections, there will be clearer alignment of results to the study aims.

Introduction

The introduction is well written and adequately sets the context for the study. Overall, there are contemporary references used however I was interested to know why the authors refer to two models that seem quite old - the Technology Acceptance Model (TAM) – reference 27 (1989) and the Unified Theory of Acceptance and Use of Technology (UTAUT) – reference 28 (2003). Are there no more recent models about acceptance of information technologies given this is an area that is growing rapidly?

In this section there are a few words that are hyphenated that do not need to be. This includes: fami-lies (line – 63), ill-ness (line – 65), deter-mines (line – 70), symp-toms (line – 84), al-most (line – 116).

Line 106 needs a full stop at the end of the sentence.

The sentence on lines 112-114 could be better written. Here is a suggestion – see changes in bold. This could support patients in evaluating and monitoring their symptoms, provide health care practitioners with a more comprehensive picture and help to detect critical symptoms in a more timely manner.

Materials and Methods

The methods are clearly presented. Under Recruitment and sampling – it could be added that a convenience sampling approach was used as that appears to be the case (mentioned under limitations on line 714). Explain why a convenience sampling approach was used in the methods section.

Under data collection – how was the interview guide developed? Was this based on a review of the literature, expert opinion etc? Were the questions piloted and if so, did they need to be changed?

I note that information regarding ethical approval for the study is provided on lines 760-762. It is preferable to include this in the main body of the paper at the end of the Methods section.

Line 165 – add the word sheet after information – i.e information sheet.

Results

Overall, the results are interesting and well presented – just a few edits required.

Table 2 – the percentage of male and female respondents can be rounded to 67% and 33%.

Line 263 – a d needs to be added to the word occure – i.e. occurred.

Lines 391-392 – this sentence needs refining.

Line 434 – what is meant by the “central evaluation criterion”?

Line 455 – delete “a” at the start of that line.

Line 467 – delete “A” at the start of the sentence.

Line 492 – delete “a” before the word monitoring.

Line 535 – between the words despite and generally add “being” – i.e. despite being generally…

Line 559 – delete “A” at the start of the sentence.

Line 586 – delete the a between “and” and “prevention”.

Discussion

A good discussion provided. A few corrections required.

Line 606 – “were considered to be treated” does not sound correct – should it be… “or were considering treatment with immunotherapy”….

Line 663 – change fac-tor to factor.

Line 678 – change be-came to became.

Line 679 – change reducing to reduce.

Line 714 – I suggest deleting “at least”.

Line 721 – change inter-view to interview.

Line 727 – change the word “surveying” as it sounds as though a survey was conducted instead of interviews.

Conclusion

Line 737 – no need to write “it turned out that.”  Just write “Overall, lung cancer patients, as potential users of digital and sensor-based monitoring systems, despite all reservations, do see potential benefits for themselves and others under certain circum-739 stances and conditions.

References

Reference 6 needs correcting.

Author Response

Please see the attachment (our answers to your comments in red)

Reviewer 2 Report

To the aoutors, 

The group is small , since the intervention took max 120 min, you could involve more patients. 

It would be nice to see educational differences as well as living areas differences( urban or rulal).

The results are resented in rather in descriptive model, some stanadrised  data  ( clear anwsers on numerical scale) would be nice to see.

There are some spellnig mistakes in the graph. Some point start with small letter some with capital letter. 

Overall it is nice concept , but rather as a preliminary study.
